# Update on Early-Life T Cells: Impact on Oral Rotavirus Vaccines

**DOI:** 10.3390/v16060818

**Published:** 2024-05-22

**Authors:** Catherine Montenegro, Federico Perdomo-Celis, Manuel A. Franco

**Affiliations:** Instituto de Genética Humana, Facultad de Medicina, Pontificia Universidad Javeriana, Bogotá 110221, Colombia; ca.montenegro@javeriana.edu.co (C.M.); perdomo_federico@javeriana.edu.co (F.P.-C.)

**Keywords:** rotavirus, vaccine, T cell, regulatory T cell, early life, layered immunity

## Abstract

Rotavirus infection continues to be a significant public health problem in developing countries, despite the availability of several vaccines. The efficacy of oral rotavirus vaccines in young children may be affected by significant immunological differences between individuals in early life and adults. Therefore, understanding the dynamics of early-life systemic and mucosal immune responses and the factors that affect them is essential to improve the current rotavirus vaccines and develop the next generation of mucosal vaccines. This review focuses on the advances in T-cell development during early life in mice and humans, discussing how immune homeostasis and response to pathogens is established in this period compared to adults. Finally, the review explores how this knowledge of early-life T-cell immunity could be utilized to enhance current and novel rotavirus vaccines.

## 1. Introduction

Despite having a very important impact on human health, oral rotavirus (RV) vaccines need improvement [1], especially in developing countries where they underperform and where RV diarrhea represents a high burden of disease [2,3]. In these settings, protective immune responses against pathogens are influenced, amongst other factors, by malnutrition and chronic gut inflammation (associated with microbial dysbiosis) that manifest as environmental enteric dysfunction (EED). In turn, EED is thought to affect oral vaccine immunogenicity and efficacy, and we currently do not have validated EED biomarkers [4,5,6,7,8]. In addition, the heterogeneity in the results of EED studies in children receiving oral RV vaccines highlights the need for more preclinical animal studies to support clinical research [8]. Oral RV vaccines face an important challenge related to the age of the intended recipients they must protect (infants and young children). In adult mice, CD8^+^ T cells are important in mediating short-term protection against reinfection, while antibodies (whose production largely depends on CD4^+^ T-cell help) mediate long-term protection [9]. Thus, T cells are key players in antiviral protection. Very little is known about immunity and T-cell responses to RV in early-life mice, particularly in local intestinal responses [10,11], and recent advances in this area suggest that early-life T cells are different from adult T cells [12,13]. Here, we will review recent advances in early-life T-cell responses in mice (a tractable model in which most T-cell studies have been performed) and humans that may impact the improvement or development of new RV vaccines. We will focus on T cells expressing the αβ antigen receptor (TCR), both effector and regulatory (Treg) cells, and, when available, we will highlight specific aspects of mucosal/intestinal T cells in this age group. We will refer to the neonatal and infant periods in humans, while we will use the term early life to refer to the neonatal and pre-weaning periods in mice.

Excellent recent reviews on T-cell responses to RV [14], immunity to intestinal viruses including RV [15], immunity in neonates [16,17], CD8^+^ T-cell immunity in early life [12], intestinal immunity in early life [13], and intestinal Treg cells [18] have recently been published. Here, we will review recent advances in the three aspects of T-cell development that are shared by both mice and humans: (a) “Layered immunity” and “neonatal window of opportunity” theories that aim to explain how immune homeostasis is established in early life (Figure 1); (b) the critical role of Treg cells to maintain tolerance to dietary and microbial components; (c) the phenotypic and functional differences in early-life effector T cells relative to adult cells. We will focus on studies that may have an impact on our understanding of T-cell responses to RV vaccines.

## 2. Early-Life T Cells in Mice

### 2.1. Layered T-Cell Immunity in Early-Life Mice

Neonates and infants suffer more infections compared to adults [13], and have poorer responses to vaccines [21], suggesting they have an impaired ability to develop long-lasting protective immunity. Historically, it was assumed that this was due to an “immature” immune system. However, recent studies suggest that neonates can develop robust immune responses in epithelial barriers to tolerate food antigens, to permit colonization by commensal microorganisms, or for pathogen elimination. However, early-life T cells have a lower capacity to generate long-term memory compared to adult T cells [13,22,23]. These functions adapted to early life are thought to be mediated by multiple populations of immune cells that appear progressively as layers with different phenotypes and functions according to the environments and challenges to which they must adapt [17,19,24]. The existence of layers of immune cells like B cells [25] and γδ T cells [26] has been known for some time but has only recently become appreciated for αβ TCR CD8^+^ T cells [12,24] and Treg cells [18,27] (Figure 1). Although the concept of a specific lineage of early-life CD8^+^ T cells is not completely clear [28], in CD8^+^ T cells (analogous to B cells [25]), the expression of Lyn28b (the master regulator of fetal lymphopoiesis [29]) is associated with features of early-life CD8^+^ T cells in mice [30]. In addition, these cells are prone to respond in a bystander, non-antigen-specific manner [28]. In humans, Lyn28b was also associated with the development of early-life Treg cells [31]. An important feature of the immune cell layers is that they can persist and modulate the response in adulthood [12]. Thus, intrinsic and environmental factors that occur in early life determine a “window of opportunity” that may influence tolerogenic versus effector-type responses that may persist and determine immune responses in the adult.

### 2.2. Mouse Treg Cells in Early Life

Early-life CD4^+^ T cells have a propensity to develop into Treg cells, and these account for the tolerogenic milieu necessary for self-discrimination and sensing of commensal microbiota [12,13]. Broadly speaking, two major Treg cell subsets can be identified in mice: those that develop in the thymus (tTreg) and those that develop in the periphery (pTreg). These subsets are difficult to differentiate phenotypically. However, tTreg cells mainly express the transcription factor Helios, while pTreg cells express the transcription factor RORγ [18]. RORγ^+^ Treg cells are strongly modulated by gut bacteria, mediate tolerance to commensals, are partially dependent on FoxP3, and are predominant in the colon relative to the small intestine. In contrast, Helios^+^ Treg cells require FoxP3, are not modulated by microorganisms, and are dominant in the small intestine [18]. The following three phases of early-life tolerance induction and development of Treg cells have been proposed [20]:

1. Neonatal phase (from 0 to 10 days of life): During this period, gut antigens and bacteria are poorly translocated into the small intestine or colon, and the development of self-tolerance probably relies on tTreg cells. Thymectomy performed during the first 2–3 days of life (but not later) in mice leads to overt multi-organ autoimmunity [27]. Self-tolerance generated at this period is mediated by a subset of Treg cells that develop in the thymus and remain in adult mice [18,27]. These Treg cells have a distinct transcriptome, activation profile, and TCR repertoire compared to adult cells [27]. Moreover, they may also play an active role in suppressing effector responses to pathogens such as herpes simplex virus more efficiently than adult Tregs [32].

2. The pre-weaning period (from 10 to 21 days of life), during which, bacteria and luminal antigens transiently traverse the intestinal epithelium. In this phase, pTreg cells (RORγ^+^) begin to generate, and they are responsible for tolerance to gut bacteria [20]. If bacteria are not encountered during this period, antigen-specific effector responses will occur in inflammatory settings [20]. During the pre-weaning period, the generation of Treg cells that mediate tolerance to the microbiota is influenced by coordinated internal and external factors in the mouse pup. The principal external factor is maternal milk: during this period, maternal milk decreases the concentration of epidermal growth factor that indirectly regulates the capacity of bacteria to traverse the colon and induce pTreg cells [20,33]. Maternal milk antibodies produced by B cells stimulated in the intestine that have migrated to the breast recognize specific subsets of bacteria and are key in the development of Treg cells, that in turn are linked to the frequency of naive (CD44^−^) intestinal T cells [23,34]. In addition, while the proportion of gut RORγ^+^ versus Helios^+^ Treg cells is determined genetically, it is also influenced by maternal antibodies, and this effect can be transmitted through multiple generations of mice [18,35]. Another factor regulating the development of RORγ^+^ Treg cells is the interplay with intestinal IgA-producing B cells during weaning. Indeed, RORγ^+^ Treg cells and IgA-producing B cells have a reciprocal regulation: RORγ^+^ Treg cells inhibit the development of intestinal IgA-producing B cells and thus the coating of the microbiota by secretory IgA. In turn, the secretory IgA coating of intestinal bacteria inhibits the development of RORγ^+^ Treg cells [35]. Mammary gland IgA is produced by cells that originally develop in the intestine, thus generating a feedback loop between RORγ^+^ Treg cells and the mother’s IgA-producing B cells specific for intestinal bacteria [35].

Synchronously with these external factors in mouse pups, during the pre-weaning period, a specialized subset of antigen-presenting cells (denominated Thetis cells) appears in the colon. A subset of these cells (of unknown function) resembles thymic antigen-presenting cells, as the cells also express the transcription factor autoimmune regulator (AIRE) [36]. Another subset of Thetis cells expresses the integrin αvβ8, which, together with αvβ6, activates latent extracellular transforming growth factor (TGF)-β. Importantly, the induction of pTreg cells by Thetis cells via TGF-β signaling is a critical process to avoid autoimmunity and colitis [36]. Other populations of tolerogenic antigen-presenting cells may also participate in the process [13]. Finally, a critical event that occurs during the pre-weaning period is the appearance of mature microfold (M) cells in gut-associated lymphoid tissues, which facilitates antigen transport and the promotion of pTreg cells [13].

3. The post-weaning phase (day 21 of life onwards) is the phase in which the translocation of intestinal bacteria and the capacity to develop tolerance to gut microbiota is stopped. After weaning, mice start to eat solid foods, and this event promotes the diversification of the microbiota, maturation of the mucosal immune cell composition, and the formation of germinal centers in lymphoid tissues [13].

The role of microbiota in the development of tolerogenic versus effector responses has important implications in the settings of enteric viral infections and EED. For instance, antibiotic microbiota ablation in mothers and their mouse pups resulted in reduced homologous RV infection/diarrhea in the pups, which was associated with a more pronounced antiviral antibody response [37]. However, a delicate balance must be maintained between commensal and non-commensal bacteria in the gut. In keeping with this notion, antibiotic exposure in early-life mice induces intestinal microbiota alterations (dysbiosis), reducing antibody responses to five parenteral human vaccines. This vaccine hyporesponse can be rescued by the restoration of commensal microbiota, but this effect is not seen in adult mice [38]. In addition, the mice that received antibiotics in early life had enhanced T-cell cytokine recall responses in vitro [38]. Thus, the level and composition of gut microbiota in early life influence the quality of antigen-specific adaptive immune responses, likely related to changes in Treg cell populations.

Malnutrition and the dysbiosis of microbiota are associated with EED, which could be associated with the reduced immunogenicity of oral RV vaccines in infants [4,5,6,7]. Recently, an EED mouse model was developed by colonizing malnourished 21-day-old mice with a single adherent-invasive *E. coli* isolate [39]. When these mice are vaccinated orally with *E. coli*-labile toxin, antigen-specific CD4^+^ T-cell responses are dampened in the small intestine but not the mesenteric lymph nodes, and vaccine efficiency is reduced. Moreover, EED mice exhibit increased frequencies of small-intestine RORγt^+^ FoxP3^+^ Treg cells. The deletion of this Treg subset restores small-intestine CD4^+^ T-cell responses and the capacity of the vaccine to protect from challenge. However, the removal of Treg cells results in increased EED-related stunting, suggesting a fine equilibrium between the capacity of the organism to thrive and defend itself from microorganisms. This model illustrates that the decreased efficacy of oral vaccines in EED may depend on RORγt^+^ FoxP3^+^ Treg cells, an effect that is restricted to local but not systemic immunity [39].

### 2.3. Effector/Memory T Cells in Early Life

Early-life effector memory CD4^+^ and CD8^+^ T cells differ from those of adults in phenotype, function, and TCR repertoire [17]. Mouse T cells begin to express the terminal deoxynucleotidyl transferase enzyme (that augments TCR variability) at one week of age, making neonatal TCRs shorter and more cross-reactive than those of adults [12]. Moreover, they express more innate Toll-like receptors (TLRs) and have increased bystander activation by innate cytokines such as interleukin (IL)-12 and IL-18 [17,28]. Consistently, early-life effector T cells have a greater capacity to proliferate to pathogens compared to adult cells [22,40,41]. However, compared to adult mice, neonatal mice have a reduced capacity to generate long-lived memory T-cell responses [12].

In the case of CD4^+^ T cells, enhanced TCR-mediated signaling enables them to respond to low antigen doses [41]. Moreover, early-life CD4^+^ T cells have a propensity to develop into Treg or Th2 cells [12]. In contrast, early-life CD8^+^ T cells can mount fast, short-lived effector responses [22]. As mentioned above, neonatal CD8^+^ T cells express Lyn28b and persist and modulate immune responses in adults [24,30]. In addition, a population of naïve (non-antigen-experienced) cells that express some memory markers (CD122^+^ CD44^+^ CD49d^−^) has been characterized in early life, and these cells also persist in adult mice and modulate responses to pathogens [24].

In addition, metabolism is an important regulator of T-cell function and differentiation. It is known that, upon activation, naïve T cells switch their metabolism infection from oxidative phosphorylation to glycolysis in response to infection to mobilize their transcriptional and translational machinery and to undergo clonal expansion. However, when infection is cleared, T cells must decrease anabolic activity to become a more quiescent memory cell, switching from glycolysis back to fatty acid oxidation [42]. Nonetheless, it was demonstrated that neonatal CD8^+^ T cells are biased to exhibit higher glycolytic activity than their adult counterparts after infection, which limits the formation of memory cells [43]. Most likely, other metabolic pathways may be implicated in the regulation of neonatal T-cell responses, which could be promising targets to improve memory formation in early life [44].

T-cell development in the intestinal mucosa differs from that in peripheral blood and non-mucosal lymphoid organs and is thought to be coordinated with the abovementioned weaning factors [13]. Very few studies have addressed the development of intestinal T cells in early-life mice [23,45,46]. At birth, intestinal T cells are infrequent, and their differential isolation from intestinal lamina propria and intraepithelial compartments is a technical challenge [23]. Thus, we do not have a clear picture of the frequencies and numbers of CD4^+^ and CD8^+^ T cells in these individual compartments in early life [23]. Before weaning, TCR αβ^+^ cells seem to be enriched in CD4^+^ over CD8αβ^+^ cells in the small intestine (evaluated as a whole) [23]. In addition, T cells in lamina propria, and to some degree in the intraepithelial compartment, have low expression of typical markers of tissue-resident memory T cells (T_RM_), like CD69. Indeed, this marker is only fully acquired post-weaning [45]. The expression of the memory marker CD44 is also reduced in lamina propria CD4^+^ T cells [23,34], but it is increased in CD8^+^ T cells due to the presence of atypical naïve cells [24].

In general, two types of effector/memory T-cell subsets exist, circulating T cells and non-circulating T_RM_ cells, that are difficult to differentiate phenotypically [47]. As an alternative, the two cell subsets are differentiated using in vivo intravascular staining (T_RM_ cells are not stained by the intravascular antibody) [47,48]. Importantly, T_RM_ cells are critical in the mucosal response against several types of pathogens [47,48]. However, while multiple studies of antiviral intestinal T cells [15,49] and studies of intestinal T_RM_ cells with model microorganisms have revealed a protective role of these subsets in mucosal tissues [50,51], very few studies have explored the role of intestinal T_RM_ cells specific for a natural intestinal pathogen. In this regard, experiments in adult mice showed that intestinal CD8^+^ T_RM_ cells modulate norovirus persistence [52]. In addition, compared to intestinal RV-specific T cells of adult mice, those of neonatal mice differ in kinetics and fine specificity, while both have a relatively short persistence [53]. Furthermore, previous studies in neonatal mouse T cells have shown that these cells may play a role in protection after RV vaccination [10] and that Treg cells expressing the latency-associated peptide (precursor of TGF-β) may modulate vaccine-induced protection [11]. The formal characterization of intestinal RV-specific T_RM_ cells in neonatal mice and their capacity to mediate protection has not been performed.

Currently, early-life mucosal T_RM_ responses have only been evaluated in the lung. As such, it has been described that, compared to adult mice, neonatal mice respond to respiratory syncytial virus (RSV) infection with variable levels of CD8^+^ T_RM_ cells (intravascular negative cells) in the lungs. In addition, neonatal mice fail to maintain lung CD8^+^ T_RM_ cells at 40 days post-infection and are less protected upon viral rechallenge [54]. However, if neonatal mice are primed and boosted with RSV in the presence of CpG (TLR9 ligand), an adult-like induction of CD8^+^ T_RM_ cells is observed, and protection is established [54]. In a second model, it was observed that, after influenza virus infection, two-week-old mice developed virus-specific CD4^+^ and CD8^+^ effector responses comparable to those in adult mice [40]. However, six weeks after primary influenza infection, the frequencies of lung virus-specific T_RM_ cells were reduced in neonatal mice relative to adults, and they were less protected upon viral challenge [40]. Interestingly, the propensity to generate fewer lung T_RM_ cell responses in early-life mice was associated with an increased expression of the transcription factor T-bet (associated with effector-like T-cell responses), and the reduction in T-bet levels in infant mice increased lung T_RM_ development [40]. These data are in line with the tendency of neonatal T cells to develop short-lived effector responses, which comes at the expense of the generation of long-term memory. In addition, these studies indicate that the limitation to establishing protective T_RM_ cells in neonates can be overcome by augmenting both innate immune activation and antigen exposure, as well as by modulating T-bet expression.

## 3. T Cells in Human Neonates and Infants

Given the difficulty of sampling blood and tissues from neonatal humans and infants, our knowledge of T-cell dynamics in early life has been scarce. However, recent studies using samples from pediatric and adult tissues have revealed how age, location, and several other variables shape T-cell function [13,48]. In humans, the neonatal period goes up to the first 28 days of life. Although there are no equivalent stages between neonatal mice and humans, and important anatomical, physiological, and microbial differences are found between both species [13], they do share some mechanisms of immune development. In this section, we will discuss key insights from neonatal and infant T-cell responses in humans.

Similar to mouse T cells, neonatal human T cells have intrinsic properties that evolved to exert a specialized role for the host. As previously discussed, the layered immune system hypothesis postulates that the temporal emergence of hematopoietic stem cells (HSCs) gives rise to diverse cell populations at different stages of life [19]. Supporting this hypothesis in humans, it has been shown that fetal CD4^+^ T cells are functionally and transcriptionally different from adult cells [55,56,57]. Specifically, fetal cells are prone to proliferation and preferentially become Treg cells [55]. In addition, fetal and adult HSCs also have different gene signatures that determine the matured T-cell profile [56]. Interestingly, preliminary findings indicate that T-cell tolerance induced in utero may be maintained until early adulthood through the establishment of long-lived Treg cells [55]. More recently, a transcriptomic analysis of naive CD4^+^ T cells revealed that cells derived from cord blood are closely related to fetal cells but distinct from circulating adult cells [58]. Moreover, cord blood cells were enriched in genes associated with a rapid proliferative response [58]. In line with these data, the single-cell transcriptomic analysis of naïve-phenotype CD8^+^ T cells from fetal, neonatal, and adult individuals revealed that multiple and distinct innate-like fetal clusters are present at birth, and some of them disappear with age [28]. These findings suggest that the human T-cell compartment is highly diverse and evolves with age. In contrast, it has been proposed that T-cell composition may progressively change from fetal-like properties towards a more adult phenotype with age. This “gradual change model” is supported by the recent single-cell transcriptomic profiling of fetal, neonatal, and adult human T cells [59]. As such, newborn cells appeared to be relatively homogeneous, and their developmental stage was placed intermediate between fetal and adult cells. Moreover, the authors propose that this divergence between the three life stages was not explained by distinct waves of HSC progenitors [59].

Relative to adults, increased frequencies of human Treg cells are found in fetal blood and lymphoid tissues, and they play an important role in suppressing effector CD4^+^ and CD8^+^ T-cell proliferation and cytokine secretion [60]. Importantly, within the first two years of life, this 6-to-10-fold higher frequency of Treg cells is also observed in multiple lymphoid and mucosal tissues (including the gut), compared with adult tissues [61], consistent with the maintenance of a tolerogenic environment during early human life. In addition, in infant blood and most of the tissues, the predominant T-cell subset exhibits a naïve CD31^+^ phenotype, consistent with recent thymic emigrants [61]. In contrast, effector memory T cells can only be found in the lungs and intestine [61]. These data illustrate the in situ control of immune responses by regulatory mechanisms in early life. In addition, the intestinal mucosa, with lower Treg/effector cell ratios, seems to constitute a hotspot of effector immune surveillance for the rapid response against exogenous antigens.

In addition to varying grades of diversity, similar to what occurs in mice [16], neonatal human T cells are biased toward broadly cross-reactive (and self-reactive) TCRs. As such, human naïve T cells in cord blood exhibit higher CD5 expression than adult cells, a marker associated with the strength of self-peptide major histocompatibility complex reactivity [62]. In line with this, T cells specific for leukemia-associated self-antigens are found in higher frequency in cord blood than in adult blood [63]. This TCR cross-reactivity might facilitate a more rapid response against different types of antigens during early life. Importantly, in addition to TCR-mediated stimulation, neonatal human T cells express several pattern recognition receptors, including TLR 2, 3, and 5, as well as complement receptors, which induce their activation, proliferation, and cytokine production [64,65]. In keeping with these observations, naïve CD8^+^ T cells from human neonates have a transcriptional and chromatin landscape that predispose them to innate-like functions [66], such as the production of interleukin-8 (CXCL8) [67]. Overall, these characteristics of neonatal T cells are reminiscent of other innate-like populations, such as B1 cells [25] and γδ T cells [26].

The above studies indicate that neonatal T cells are armed with innate immune features that could contribute to a rapid host defense upon infection. Nonetheless, a consequence of this lower activation threshold in human neonatal T cells is increased proliferation and more rapid differentiation towards short-lived cells [40,41,68], as has been described for neonatal mice cells [22]. These intrinsic properties could affect the development of long-term memory, with important consequences for secondary immune responses. On the one hand, it has been described that this different program of human newborn T cells facilitates Treg cell differentiation [31]. In addition, it was shown that CXCL8^+^ CD4^+^ T cells from neonates are direct precursors of Th1 cells upon sustained proliferation [69]. On the other hand, recent studies have demonstrated that, similar to mouse cells, human infant T cells are intrinsically programmed for short-term responses, driven by the increased expression of the transcription factor T-bet [40] and other transcription factors [28,70]. As such, infant cells exhibit higher levels of T-bet relative to adult cells in vivo and after in vitro activation, which negatively correlates with the expression of the long-lived memory marker CD127 [40]. In agreement with these findings, CD8^+^ T cell responses of children (less than 4 years old) to an inactivated vaccine at day 10 after vaccination are similar to those of adults; however, contrary to adults, the response had practically disappeared by day 28 after vaccination [71]. Also, in keeping with these data, a recent study reported that children maintain a novel CD8^+^ T cell subset epigenetically poised for rapid effector responses, and this subset is lost with age [72]. Of note, the predisposition for effector-like responses in the neonatal period also impacts the generation of CD69^+^ CD103^+^ T_RM_ cells in the respiratory tract. Indeed, the frequency of this tissue-resident population increases with age, suggesting that the aforementioned transcriptional program of neonatal T cells directly regulates the generation of this memory population [73]. A similar dynamic seems to occur for T_RM_ cells in the human neonatal gut [74]. In the early weeks of human life, there is a low frequency of αβ T_RM_-phenotype cells in the gut, and this population accumulates progressively with age, peaking at around 1.4 years and remaining stable thereafter [75,76].

In summary, neonatal human T cells are intrinsically programmed for rapid differentiation to effector-like responses. Enhanced TCR signaling and localized effector-like responses could protect newborns from newly encountered pathogens. However, a higher proportion of effector-phenotype T cells in infants could contribute to tissue injury during viral infection [77]. Moreover, short-lived effector responses in neonates are generated at the expense of the induction of long-term memory. As a result, the generation of T_RM_ cells at the site of infection during early life may undermine the immune response, resulting in recurrent symptomatic disease. Likely, the promotion of localized effector-like immune responses, such as in bronchus- and gut-associated lymphoid tissues [78], could be a beneficial vaccine strategy in early life to prevent infection or illness, waiting for the developmental maturation and acquisition of adult immune traits.

## 4. Conclusions and Future Directions: How May Early-Life T-Cell Immunity Impact RV Vaccines?

Recent studies have provided important insights into the influence of the neonatal period on the establishment of enduring immune responses and the implications for health and disease. Beyond the inherent attributes of mouse and human T cells discussed above, various additional elements modulate and shape the neonatal immune system. Within this “neonatal window of opportunity”, environmental factors, such as commensal microbiota or some specific pathogens, may shift immune responses towards more tolerizing or effector-type responses and thus predispose an individual to immune-mediated and other diseases in adult life [79]. For instance, it has recently been shown that early microbial exposure modulates and programs mouse CD8^+^ T-cell development, and fetal-derived cells in a dirty environment are more responsive to stimulation, acquiring an effector-like profile [80]. Considering that RV vaccines provide non-sterilizing immunity in children and they are less effective in low-income countries [9], it is possible that poor environmental conditions in these settings influence early-life immunity. A potential outcome of these processes is the generation of monofunctional effector-like responses [81] and low frequencies of long-lived memory RV-specific cells [82], thus impairing long-term immunity in adult life [83]. Likely, poor tissue-resident memory responses are also affected.

As mentioned previously, protection against RV reinfection in adult mice is mediated by CD4^+^ T cell-dependent antibodies and by short-lived CD8^+^ T cells [9], and the role of early-life T cells is much less well known. The following studies of early-life T cells may be envisioned for preclinical basic research or clinical studies that could have an impact on the improvement/development of new RV vaccines:

1. Promotion of T_RM_-cell responses: The improvement of antigen delivery to the gut tissue by oral vaccines [78] or using parenteral vaccines to boost after priming with an oral vaccine could be beneficial strategies in early life.

2. Innate immune activation simultaneously with mucosal antigen administration [54]. Double-stranded RNA and dmLT are two mucosal adjuvants that are being tested in ongoing clinical trials with orally administered vaccines [84].

3. The modulation of transcriptional circuits to promote long-lived memory T cells in early life, like reducing T-bet expression [40]. Moreover, other transcriptional or metabolic pathways could be modulated to promote memory-like cells. In this regard, previous studies have shown that inhibition of the mTORC1 (mammalian target of rapamycin complex 1) pathway promotes memory T-cell differentiation in mice [85], as well as enhances immune function and the response to influenza vaccination while reducing infections in the elderly [86].

4. The study of Treg cells in EED and RV-vaccinated children [39]. In this regard, since the evaluation of RORγt^+^ Treg cells (or their elimination seeking a benefit) is difficult given the need for tissue isolation [35], measuring antibody-coated bacteria in stool samples could be a more accessible and useful correlate of Treg cells [85], as well as a potential biomarker in children [87]. Given the impact of maternal milk on the thymus and T cells in general [88], these studies would need to be performed with breast-fed and formula-fed children as separate groups, but in both groups, levels of IgA-covered bacteria may correlate with Treg cells.

5. Phenotypic and functional innate-like features of T cells may be used as biomarkers of anti-RV immune responses and vaccine responses.

6. Our lack of understanding of basic aspects of early-life T cells highlights our need for more basic studies in neonatal mouse models of RV infection/vaccination [11]. As observed in this review (and summarized in Table 1), several basic aspects of the early-life immune response of mice also seem to occur in human neonates and infants, justifying these studies.

However, some important differences between human and mouse early-life T cells exist; for example, the diversification of the T-cell repertoire occurs earlier in humans than in mice and the post-thymic maturation of T cells is shorter in mice [17]. For this reason, the extrapolation of results from mice to humans should be carried out with caution.

In conclusion, we are just starting to understand how the gradual diversification of the T-cell compartment and functions in early life may determine the clinical outcome after infection. To improve RV vaccines, it is crucial to consider environmental factors such as diet, microbiome, and the level of exogenous antigen exposure, which can alter T-cell immune ontogeny, along with the regulatory mechanisms that govern this layered immunity. The use of advanced molecular techniques for global immune monitoring could provide valuable tools for developing next-generation RV vaccines. These vaccines should not only target antigen-specific responses but also promote effective tissue-localized and long-lived memory responses.

## Figures and Tables

**Figure 1 viruses-16-00818-f001:**
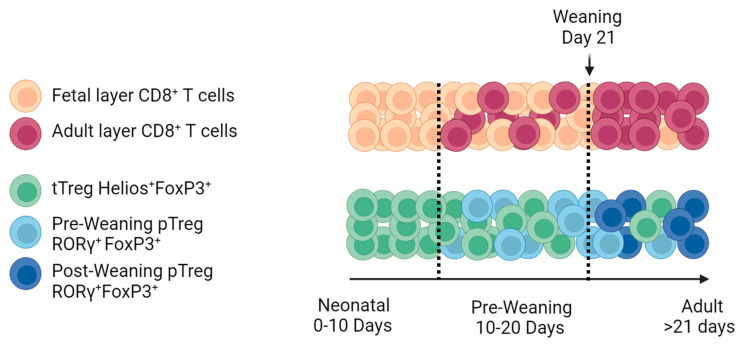
Layered development of CD8^+^ T cells and CD4^+^ Treg cells in early-life mice. Many cells of the immune system appear progressively as layers of cells with distinct functions [19]. A small fraction of each layer persists in adults [19]. Recently, this has been shown to occur for early-life CD8^+^ T cells, which express Lyn28b, the master regulator of fetal lymphopoiesis [12]. Particularly, early-life CD8^+^ T cells are characterized by bystander activation by innate cytokines and the acquisition of an effector-like profile. On the other hand, in early life, CD4^+^ T cells are biased towards the development of Treg cells that also seem to develop in layers [18]. The first layer of Treg cells to appear seems to be enriched in Helios^+^ thymus-derived Treg (tTreg) cells that are mostly selected to mediate self-tolerance [18]. Between day 10 of life and the moment of weaning (day 21 of life), a second layer of RORγ^+^ peripheral Treg (pTreg) cells develops, and they mediate tolerance mostly to gut bacteria [20]. Figure designed with Biorender.

**Table 1 viruses-16-00818-t001:** Summary of characteristics shared by human and mouse early-life T cells.

T cells are functionally and transcriptionally different from adult T cells, supporting the hypothesis of the layered immunity theory.
T cells exhibit increased expression levels of innate-like receptors and respond in a bystander fashion.
TCRs are cross-reactive/autoreactive with limited diversity due to a lack of expression of TdT.
CD8^+^ T cells are prone to rapidly proliferating and differentiating in short-lived cells depending on the expression of T-bet.
Long-lived memory T-cell responses are diminished.
CD4^+^ T cells are prone to become Treg cells, and both humans and mice have comparable Helios^+^ and RORγ^+^ Treg subsets.
Both seem to have reduced capacity for the generation of mucosal T_RM_.

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
