# Peer review of "Update on Early-Life T Cells: Impact on Oral Rotavirus Vaccines"

_viruses, 2024, doi:10.3390/v16060818_

Round 1

Reviewer 1 Report

Comments and Suggestions for Authors

Review of “Update on early life T cells: impact on rotavirus vaccines” by Montenegro, Perdomo-Celis, and Franco

This is a timely review on a key emergent aspect of vaccine-induced immunity in neonatal and early life by a group that has advanced our understanding of rotavirus immunity considerably. Although not unequivocally established, the layered immune system hypothesis (Herzenberg and Herzenberg, 1989) posits that HSPCs giving rise to a tolerogenic fetal immune system are superseded by distinct HSPCs that instead give rise to a protective adult HSPC-derived immune system and that for a time these immune systems are layered upon each other, exerting opposing influences. The alternate possibility is that the fetal-to-adult transition might instead occur through relatively uniform progressive maturational changes across immune cell populations, leading to a continuous spectrum of intermediate phenotypes (supported by some single cell transcriptome studies). Of course, these two explanations are not necessarily mutually exclusive and a thorough review of the emerging model in the context of rotavirus vaccines is important. The importance of metabolism (through gut microbiota and nutrition in early life) in shaping T cell functions through the fetal-to-adult transition is also a key finding for this field and is reviewed by Montenegro and colleagues. 

Overall, this review is well-written and relevant publications have been cited. The sole figure provided is useful and the authors should consider adding another visual/graphic at the end that summarizes the conclusions and future directions proposed (the future directions were among the most interesting take away messages for me from the review). 

Minor suggestions and comments are given below:

- The role of metabolic reprogramming in the activation of peripheral T cells has been relatively well-delineated in the literature. However, its role in early T cell development in the thymus is just surfacing and may be worth mentioning. Since both nutrition and gut microbiota have a major effect on the metabolic profile of T cells, some comments on metabolic states (or differences in regulators of metabolic states such as mTORC1/2, epigenetic histone modifications etc) uncovered in early life vs. adult T cells would be useful to readers. 

- The authors could add some discussion (in future directions) on whether the presence of innate-like and tolerogenic features of the adaptive immune system be used as biomarkers (by epidemiological surveillance efforts for RV immune responses) to improve vaccination strategies against infections in early life?

- As neonatal CD8+ T cells have a marked innate immune response transcriptional (and protein) signature, one interesting question arising (that can be included in the review) is how this cell population is functionally relevant for rotavirus host range restriction in both the mouse and human model of natural disease, due to the use of such naturally host-attenuated strains as vaccines. 

- Overall, there are striking similarities between reprogramming of naive CD4+ T cells across age, in cancer, and in response to viral infections (quiescence versus an active state). This has not been discussed and can be mentioned since clinical seno- and onco-therapeutics can be likely repurposed for viral vaccines. 

- Lines 316-319: In addition to t-bet, transcription factors that sifferentiate neonatal and adult T cell functions include higher expression in adult cells of the master transcription factor RUNX3 (Cruz-Guilloty et al., 2009), and BCL11A and CEBPE in neonates (Yu et al 2012, Gombart & Koeffler, 2002)

- Lines 299-301: also TLR3 is higher in neonatal human T cells (see Galindo-Albarran reference cited); 

- Due to the rapid but short lived memory (and secondary immunity) of neonatal T cells, what are the implications for boosting rotavirus vaccination later in life? Are there any studies in mice that would shed some light on this question?

Two key references that can be included:

- Montecino-Rodriguez E. et al Differential Expression of PU.1 and Key T Lineage Transcription Factors Distinguishes Fetal and Adult T Cell Development. J. Immunol. 2018; 200: 2046-2056

- Li N. et al. Memory CD4+ T cells are generated in the human fetal intestine. Nat. Immunol. 2019; 20: 301-312

Reviewer 2 Report

Comments and Suggestions for Authors

This review provides a very comprehensive overview of T cell biology in neonates, which is a field that has seen some significant recent updates. The review is nicely structured and well written, with extensive referencing to the relevant early life literature. It is especially valuable to see the comparison between mouse and human T cell knowledge in section 3.

It would be good to signpost readers to a related systematic review on T cell responses to rotavirus infection and vaccination by Laban et al 2022 (Viruses). The aims of the review under consideration are clearly distinct from the aims of Laban et al’s review, but the latter provides a useful summary of previous clinical studies. This could be mentioned in the intro, and/or at the start of section 3.

Line 20 – recommend specifying that rotavirus-induced gastroenteritis represents a high burden as currently seems to refer to vaccination.

Line 34 – redefine ‘they’ to improve clarity

Line 46 – it is not clear what ‘a)’ is referring to – please briefly clarify these models/theories

Line 69 – could also mention interference by maternal antibodies as a reason for poor vaccine responses in neonates.

Line 151 – It is really interesting how antibiotic treatment of pups reduces RV infection/disease in the study referenced. This is the opposite effect to that recently reported in adult mice by Schnepf et al 2021 (Plos One). Are you able to comment on this difference?

Line 223-242 - focus here is on lung responses to pathogens in early life, so not immediately obvious how it fits with the rotavirus focus of the title. Recommend making the reason for including this section clearer to the reader.

Line 341 – suggest rephrasing question in the subtitle to ‘How may early-life T cell immunity impact RV vaccines?’

Line 361 – the current orally delivered vaccines will be inducing localized effector immune responses, please explain how strategy 1 is different, and/or how TRM specific responses might be achieved.

Line 367 – are there examples/suggestions of how this can be achieved?

Reviewer 3 Report

Comments and Suggestions for Authors

This is an informative review evaluating the early life development of T cells in mice and humans, underscoring how the regulated development of T cell function as a function of age might impact mucosal and systemic short-term and long-term immunity, with special focus on the context of rotavirus infection and oral vaccination.  I do not have any major issues with the manuscript as a whole, which in general I found to be well written, interesting, and enjoyable to read. I do think there could be a few helpful edits and additions that would add even more impact to the manuscript, as listed below, but none are critical in my opinion.

General comments:

·       Perhaps not surprisingly given the difficulty of conducting such studies in humans, the bulk of the paper focuses on the mouse model. However, a complementary figure highlighting the proposed mechanisms of human neonatal T cell development, similar to Figure 1 for the mouse, would be useful, or a figure or table that highlights both the similarities, and in particular, the differences between mouse and human, especially given the call for more neonatal mouse model development (line 375). For example, unlike in mice, neonates who undergo early thymectomy (or have thymic issues due to developmental abnormalities) do not undergo the profound auto-immunity that is seen with neonatal mouse thymectomy.

·       Has there been any work highlighting the role of human milk on T-cell development, as has been presented for mouse?

·       Are there any informative data from studies of perinatal or early-life viral infections in newborns and young infants that would further support (or argue against) the themes proposed here? Early post-natal CMV infection (not congenital infection) comes to mind here, as one of the interesting hallmarks of this infection is the inability for young infants in this setting to control virus replication, even in the absence of clinical disease, which would seem to argue against a bias for infant T cells towards rapid effector function (although this admittedly could be a gross oversimplification). Studies of live-attenuated influenza vaccine responses in young children might be another source of pertinent data.

·       The possibilities for specific applications of these themes towards enhancing rotavirus vaccine performance are interesting. However, given the starting premise that antibodies, not CD8+ T-cells, appeared to be most important for long-term immunity (lines 29-31), is the main goal to improve T-cell function and memory as a means to further augment long-term antibody production, to generate better effector T cells, or both? This could be a bit more clearly articulated in the conclusion.

·       Lines 369-373: Measurements of IgA-coated bacteria in children has been performed a number of times to date (reviewed here for example: https://www.sciencedirect.com/science/article/pii/S1931312816302694?via%3Dihub.) As written, it is not clear to me how these measurements could serve as a biomarker of Tregs in breastfed children, where a substantial amount of bound IgA may be of maternal origin.

Specific comments:

·       Lines 20-21: Suggest “…and where rotavirus diarrhea represents a high burden of disease.”  

·       Lines 160-161: I think the opening sentence of this paragraph is a bit overstated. There is no clear consensus on the role or definition of EED in humans or the role of the microbiome and how it relates to RV vaccine performance, and all the papers cited here (4-7) investigate vaccine immunogenicity via antibody seroconversion, NOT clinical efficacy.

·       Line 283: “…(within the first two years of life)”.
